

# Analytical results for a coagulation/decoagulation model on an inhomogeneous lattice

**Nicolas Crampé**

Laboratoire Charles Coulomb (L2C), UMR 5221 CNRS-Université de Montpellier,
Montpellier, France.

* nicolas.crampe@umontpellier.fr

## Abstract

We show that an inhomogeneous coagulation/decoagulation model can be mapped to a quadratic fermionic model via a Jordan-Wigner transformation. The spectrum for this inhomogeneous model is computed exactly and the spectral gap is described for some examples. We construct our inhomogeneous model from two different homogeneous models joined by one special bond (impurity). The homogeneous models we started with are the coagulation/decoagulation models studied previously using the Jordan-Wigner transformation.

*In memory of Maxime Clusel and Vladimir Lorman*



# 1 Introduction

The description of the non-equilibrium stationary state (NESS) of a macroscopic system is much less understood than the equilibrium case. One major difference is that the behavior of the NESS is essentially non-local whereas that systems at equilibrium (away from the critical point) is local. This implies that local changes of the non-equilibrium model may have a significant repercussion in physical quantities even away from this modification. It is why the study of the effects of the boundaries or the introduction of impurities in such models has attracted as much attentions.

In one-dimensional models, this behavior is even heightened. In this case, we can hope that some exact results for particular models can be obtained. For example, numerous exact results have been computed for exclusion processes where one particle moves differently from the other ones [5, 11, 12, 15, 26, 28]. Unfortunately, for stationary defect (*i.e.* the rates are modified at particular bonds), very few exact results have been computed. To the best of our knowledge, it is only for parallel dynamics and deterministic hopping that analytical results exist [20, 31]. In the case of the asymmetric simple exclusion process (ASEP) which can be solved analytically for a homogeneous lattice, the effects of a static impurity and the formation of shocks have been intensively studied by various methods [13, 16, 17, 22, 24, 29, 30, 33, 35]. Exact results have been obtained only in the low-current regime [34]. Let us also mention that the introduction of a static impurity for other integrable systems has also been studied intensively [4, 6–8, 10]: there exist strong constraints on the type of impurity and on the bulk coupling constants such that the model with the impurity remains integrable.

In this paper, we solve analytically an inhomogeneous Markovian model composed of two segments with different hopping rates. These two segments are joined by a bond whose rates are computed such that the analytical resolution remains possible. When the rates in both segments are identical, we recover a model with one impurity. It is well-known that a homogeneous coagulation/decoagulation model (see section 2.1 and figure 1) can be mapped to a free fermion model [1, 18, 19, 25]. We show in this paper that this type of mapping is still possible for an inhomogeneous model based on two different homogeneous coagulation/decoagulation models joined by a bond (see section 2.2). The techniques needed to obtain the spectrum of the homogeneous model are recalled in section 3 and are generalized to the inhomogeneous case in section 4. More precisely, we show that the spectrum of the Markov matrix is given by the roots of a polynomial (see equation (39)) of degree the length of the chain. This polynomial is expressed in terms of the Chebyshev polynomials. Finally, in section 5, two examples are worked out for which the spectral gap is computed.

# 2 Solvable Markovian models on inhomogeneous lattice

In this section, we show that we can construct a Markovian model on an inhomogeneous lattice which can be mapped to a quadratic fermionic model. Similar mappings have been obtained previously in [1, 18, 19, 25] for homogeneous lattices. We recall these results in section 2.1 for a coagulation/decoagulation model and then we generalize them to the inhomogeneous model in section 2.2.

## 2.1 Solvable model on homogeneous lattice and the quantum formalism for its master equation

We present now the master equation for the Markovian model on a homogeneous lattice of a particular coagulation/decoagulation process: the rates are chosen such that the Markovian matrix can be mapped to a quadratic fermionic model [1, 18, 19, 25].

   We consider a stochastic process which describes particles moving on a one-dimensional lattice of L sites with at most one particle per site. The time evolution is governed by the following rules. During each infinitesimal time $dt$, a particle in the bulk can jump to the left (resp. right) with probability proportional to $qdt$ (resp. $pdt$) on the neighbouring site if it is empty. If two neighbouring sites are simultaneously occupied, the left (resp. right) particles can disappear with the rate $pdt$ (resp. $qdt$). A particle can also appear on the left (resp. right) neighbouring empty site of a particle present on the lattice with the rate $\Delta qdt$ (resp. $\Delta pdt$). A summary of these rates is presented on figure 1. Let us emphasize that the parameters $p$, $q$

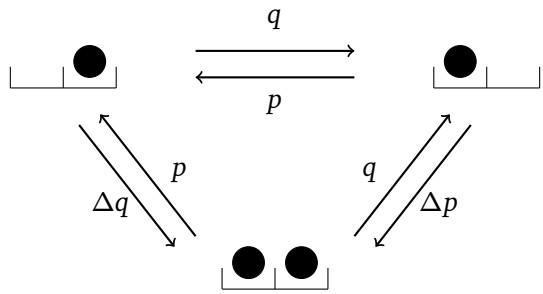

Figure 1: Non vanishing rates between the different configurations at two neighboring sites in the bulk.

and $\Delta$ are real positive numbers such that the probabilities remain positive.

   We recall now the quantum Hamiltonian formalism, which is suitable for the following computations, used to present the master equation (see [32] for details). The configurations of the previous process are in one-to-one correspondence with the vectors describing a system of $L$ spins $\frac{1}{2}$. Indeed, the spin vector $|\sigma_1, \sigma_2, \ldots, \sigma_L\rangle$ with $\sigma_i = \pm 1$ corresponds to a configuration with one particle at site $i$ if $\sigma_i = -1$ and zero particle if $\sigma_i = +1$. The probability $P_t(\sigma_1, \sigma_2, \ldots, \sigma_L)$ at time $t$ to be in the configuration $|\sigma_1, \sigma_2, \ldots, \sigma_L\rangle$ can be encompassed in the following state vector

$$P_t = \sum_{\sigma_i = \pm} P_t(\sigma_1, \sigma_2, \ldots, \sigma_L)|\sigma_1, \sigma_2, \ldots, \sigma_L\rangle. \tag{1}$$

Then the master equation describing the time evolution of the probabilities can be written as follows

$$\frac{dP_t}{dt} = MP_t, \tag{2}$$

where $M$ is the Markov matrix. For the process studied here for which only two neighbouring sites are considered at each infinitesimal time, it reads

$$M = \sum_{k=1}^{L-1} m_{k,k+1}, \tag{3}$$

where the subscripts indicate on which spins the matrix $m$ acts on non trivially. In the basis

$|+\rangle = \begin{pmatrix} 1 \\ 0 \end{pmatrix}$ and $|-\rangle = \begin{pmatrix} 0 \\ 1 \end{pmatrix}$, the local jump operator $m$ is given by

$$m = \begin{pmatrix} 0 & 0 & 0 & 0 \\ 0 & -(\Delta+1)q & p & p \\ 0 & q & -(\Delta+1)p & q \\ 0 & \Delta q & \Delta p & -p-q \end{pmatrix}. \tag{4}$$

For later convenience, we define the angle $0 \leq \theta < \pi/4$ by

$$\Delta = \tan^2(2\theta). \tag{5}$$

In order to perform the mapping to the fermionic operator, we rewrite the local jump operator $m$ as

$$m_{12} = \widetilde{m}_{12} + t(S_1^x - S_2^x) \quad \text{with} \quad \widetilde{m}_{12} = aS_1^+S_2^- + bS_1^-S_2^+ + cS_1^+S_2^+ + dS_1^-S_2^- + hS_1^z + \bar{h}S_2^z + f. \tag{6}$$

We have used the following definitions:

$$a = \frac{p\cos^4(\theta) + q\sin^4(\theta)}{\cos^2(2\theta)} \qquad b = \frac{q\cos^4(\theta) + p\sin^4(\theta)}{\cos^2(2\theta)} \qquad c = \frac{(p+q)\cos^4(\theta)}{\cos^2(2\theta)} \tag{7}$$

$$d = \frac{(p+q)\sin^4(\theta)}{\cos^2(2\theta)} \qquad h = \frac{p}{2\cos(2\theta)} \qquad \bar{h} = \frac{q}{2\cos(2\theta)} \tag{8}$$

$$t = \frac{1}{4}(p-q)\tan^2(2\theta) \qquad f = -\frac{p+q}{4}\left(1 + \frac{1}{\cos^2(2\theta)}\right) \tag{9}$$

and

$$S^+ = \sin^2(\theta)\begin{pmatrix} 1 & -\frac{\cos(2\theta)}{2\sin^2(\theta)} \\ \frac{2\sin^2(\theta)}{\cos(2\theta)} & -1 \end{pmatrix} \qquad S^- = \cos^2(\theta)\begin{pmatrix} 1 & \frac{\cos(2\theta)}{2\cos^2(\theta)} \\ -\frac{2\cos^2(\theta)}{\cos(2\theta)} & -1 \end{pmatrix} \tag{10}$$

$$S^z = \cos(2\theta)\begin{pmatrix} 1 & 1 \\ \tan^2(2\theta) & -1 \end{pmatrix} \qquad S^x = S^+ + S^- \quad \text{and} \quad S^y = i(S^- - S^+). \tag{11}$$

The previous matrices $S^\pm$, $S^x$, $S^y$ and $S^z$ are the Pauli matrices in an unusual basis. We recover the usual representation of the Pauli matrices by a simple conjugation.

By using the form (6) of $m$, the Markov matrix (3) becomes

$$M = \widetilde{M} + t(S_1^x - S_L^x) \quad \text{with} \quad \widetilde{M} = \sum_{k=1}^{L-1} \widetilde{m}_{k,k+1}. \tag{12}$$

The bulk part $\widetilde{M}$ of the Markov matrix is quadratic in terms of the Pauli matrices $S^+$ and $S^-$ (we recall that $S^z = 2S^+S^- - 1$) up to a constant term. Then, it can be mapped to a free fermionic model [27]. The boundary terms in $M$ seem problematic since they are linear in $S^+$ and $S^-$ however this problem has been overcome in [2,3,19]. Before coming back to this problem in section 3, we want to present the first new result of this paper: the construction of a Markovian model on an inhomogeneous lattice with the bulk part quadratic in terms of $S^+$ and $S^-$.

## 2.2 Inhomogeneous model equivalent to a quadratic fermionic model

In this section, we want to obtain similar Markovian model to the previous one but with the rates depending on the sites. Such an inhomogeneous model is obtained by juxtaposing two

segments with different rates and connecting them by an impurity bond: we consider a first segment from 1 to $L_1$ where the rates are given by $p_1$, $q_1$ and $\Delta_1 = \tan^2(2\theta_1)$ and a second segment of length $L_2$ from $L_1 + 1$ to $L_2 + L_1$ where the rates are given by $p_2$, $q_2$ and $\Delta_2 = \tan^2(2\theta_2)$. We want to determine the rates between the sites $L_1$ and $L_1 + 1$ such that the whole model from 1 to $L_1 + L_2$ can be transformed to a quadratic fermionic model. More precisely, we look for the $4 \times 4$ matrix $m^{\text{junc}}$ such that the following Markovian matrix

$$M = \sum_{k=1}^{L_1-1} m_{k,k+1}^{(1)} + m_{L_1,L_1+1}^{\text{junc}} + \sum_{k=L_1+1}^{L_1+L_2-1} m_{k,k+1}^{(2)}, \tag{13}$$

can be mapped to a quadratic fermionic model using a Jordan-Wigner transformation. In relation (13), the notation $m^{(i)}$ stands for the matrix $m$ given by (4) where $p$, $q$ and $\Delta$ are replaced by $p_i$, $q_i$ and $\Delta_i$. The local jump operators $m^{(1)}$ and $m^{(2)}$ can be written as in relation (6) where $S_j^\#$ (for $\# = \pm, x, y, z$) are given by (10) and (11) but with $\theta$ replaced by $\theta_1$ if $1 \leq j \leq L_1$ and by $\theta_2$ if $L_1 + 1 \leq j \leq L_1 + L_2$.

We look for $m^{\text{junc}}$ in the form

$$m^{\text{junc}} = \alpha S^+ \otimes S^- + \beta S^- \otimes S^+ + \gamma S^+ \otimes S^+ + \delta S^- \otimes S^- + \eta S^z \otimes \mathbb{I} + \bar{\eta} \mathbb{I} \otimes S^z + \psi + \tau S^x \otimes \mathbb{I} + \bar{\tau} \mathbb{I} \otimes S^x \tag{14}$$

where $\mathbb{I}$ is the $2 \times 2$ identity matrix and $S^\#$ are given by (10),(11) with $\theta$ replaced by $\theta_1$ (resp. by $\theta_2$) in the first (resp. second) space. The values of $\tau$ and $\bar{\tau}$ must be chosen such that they compensate the boundary term on the site $L_1$ coming from the first segment and the boundary term on the site $L_1 + 1$ coming from the second segment:

$$\tau = \frac{p_1 - q_1}{4} \tan^2(2\theta_1) \quad \text{and} \quad \bar{\tau} = -\frac{p_2 - q_2}{4} \tan^2(2\theta_2). \tag{15}$$

The first result of this paper consists in finding $\alpha, \beta, \gamma, \delta, \eta, \bar{\eta}$ and $\psi$ such that $m^{\text{junc}}$ be Markovian. We get that the impurity local jump operator given by:

$$m^{\text{junc}} = \begin{pmatrix} Q_2 - Q_1 & 0 & 0 & 0 \\ \bar{q}\Delta_2 - \overline{Q} - Q_2 & -\overline{Q} - Q_1 - \bar{q} & \bar{p} & \bar{p} \\ \bar{p}\Delta_1 - \overline{Q} + Q_1 & \bar{q} & -\overline{Q} + Q_2 - \bar{p} & \bar{q} \\ 2\overline{Q} - \bar{p}\Delta_1 - \bar{q}\Delta_2 & \overline{Q} + Q_1 & \overline{Q} - Q_2 & -\bar{p} - \bar{q} \end{pmatrix} \quad \text{with} \quad Q_i = \frac{\Delta_i(q_i - p_i)}{2} \tag{16}$$

can be written as (14) with

$$\alpha = \frac{\bar{p}\cos^2(\theta_1)}{\cos(2\theta_1)} - \frac{\bar{q}\sin^2(\theta_2)}{\cos(2\theta_2)} + \frac{\overline{Q}}{2} \qquad \beta = -\frac{\bar{p}\sin^2(\theta_1)}{\cos(2\theta_1)} + \frac{\bar{q}\cos^2(\theta_2)}{\cos(2\theta_2)} + \frac{\overline{Q}}{2} \tag{17}$$

$$\gamma = \frac{\bar{p}\cos^2(\theta_1)}{\cos(2\theta_1)} + \frac{\bar{q}\cos^2(\theta_2)}{\cos(2\theta_2)} + \frac{\overline{Q}}{2} \qquad \delta = -\frac{\bar{p}\sin^2(\theta_1)}{\cos(2\theta_1)} - \frac{\bar{q}\sin^2(\theta_2)}{\cos(2\theta_2)} + \frac{\overline{Q}}{2} \tag{18}$$

$$\eta = \frac{\bar{p}}{2\cos(2\theta_1)} \qquad \bar{\eta} = \frac{\bar{q}}{2\cos(2\theta_2)} \tag{19}$$

$$\psi = \frac{p_1 - q_1}{4}\tan^2(2\theta_1) + \frac{q_2 - p_2}{4}\tan^2(2\theta_2) - \frac{\overline{Q} + \bar{p} + \bar{q}}{2}. \tag{20}$$

Let us emphasize that $m^{\text{junc}}$ given by (16) is the most general Markovian matrix with this property. The parameters $\bar{p}$, $\bar{q}$, $\overline{Q}$ are new free parameters characterizing the rates on the impurity. The different processes at the impurity with their rates are displayed on figure 2. The positivity of the rates for the impurity imposes some constraints on the parameters:

$$Q_1 \geq Q_2, \qquad \bar{q} \geq 0, \qquad \bar{p} \geq 0, \tag{21}$$

$$2\overline{Q} \geq \bar{p}\Delta_1 + \bar{q}\Delta_2, \qquad \bar{p}\Delta_1 + Q_1 \geq \overline{Q} \geq -Q_1, \qquad \bar{q}\Delta_2 - Q_2 \geq \overline{Q} \geq Q_2. \tag{22}$$

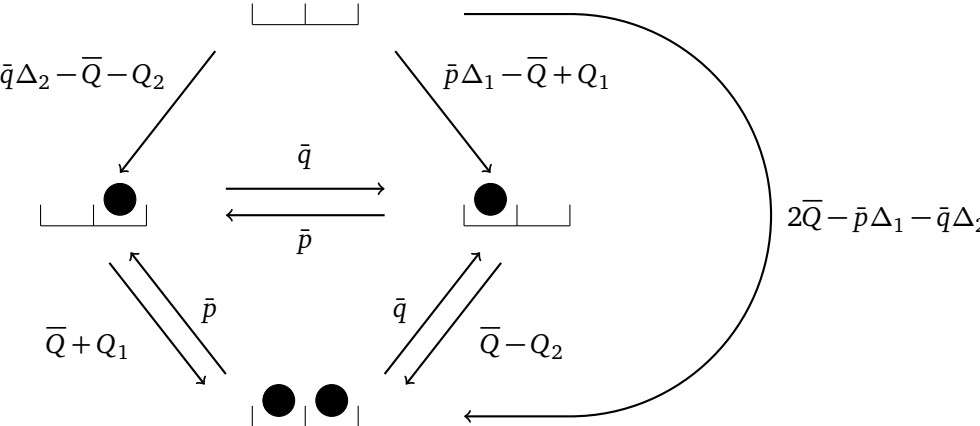

Figure 2: Non vanishing rates between the different configurations at the impurity junction. The left site on the figure corresponds to the site $L_1$ of the lattice.

Let us remark that we recover an homogeneous model of length $L_1 + L_2$ for

$$p_1 = p_2 = \bar{p} = p, \qquad q_1 = q_2 = \bar{q} = q, \qquad \theta_1 = \theta_2 = \theta \quad \text{and} \quad \overline{Q} = \frac{q+p}{2}\tan^2(2\theta). \quad (23)$$

We show in section 4 that the resolution of this inhomogeneous model is possible using a mapping to a free quadratic fermionic model. This mapping is possible since the bulk part of its Markovian matrix given by (13) and (16) are quadratic in $S^+$ and $S^-$.

## 3 Resolution of the homogeneous model

In this section, before solving the inhomogeneous model, we recall well-known results concerning the resolution of the homogeneous model given by the Markov matrix (12).

As explained previously, the matrix $\widetilde{M}$ in (12) can be mapped to a quadratic fermionic model [27] but it is not the case for $M$ due to the boundary terms which are linear in $S^x$. As explained in [2,3,19], to deal with these boundary terms, we must modify slightly the Markov matrix. We add additional sites, called 0 and $L + 1$ at both ends of the chain and define

$$M' = \widetilde{M} + tS_0^x S_1^x - tS_L^x S_{L+1}^x, \quad (24)$$

with $\widetilde{M}$ acting trivially on the sites 0 and $L + 1$. The spectrum of $M'$ decomposes into 4 sectors $(++)$, $(+-)$, $(-+)$ and $(--)$ which correspond to the eigenvalues of $(S_0^x, S_{L+1}^x)$. We recover exactly the spectrum of the Markov matrix $M$ in the sector $(++)$ [2,3,19].

Using the Jordan-Wigner transformation [23], we define the fermionic creation and annihilation operators

$$\mathfrak{a}_k^+ = S_k^+ \prod_{j=0}^{k-1} S_j^z \quad \text{and} \quad \mathfrak{a}_k^- = S_k^- \prod_{j=0}^{k-1} S_j^z, \quad (25)$$

which satisfy the canonical anticommutation relations

$$\{\mathfrak{a}_k^-, \mathfrak{a}_\ell^+\} = \delta_{k,\ell}, \qquad \{\mathfrak{a}_k^-, \mathfrak{a}_\ell^-\} = 0 \quad \text{and} \qquad \{\mathfrak{a}_k^+, \mathfrak{a}_\ell^+\} = 0. \quad (26)$$

Using this transformation, $M'$ can be expressed as a combination of $\mathfrak{a}_k^{\pm}$:

$$M' = \sum_{k=1}^{L-1} \left( -a\, \mathfrak{a}_k^+ \mathfrak{a}_{k+1}^- + b\, \mathfrak{a}_k^- \mathfrak{a}_{k+1}^+ - c\, \mathfrak{a}_k^+ \mathfrak{a}_{k+1}^+ + d\, \mathfrak{a}_k^- \mathfrak{a}_{k+1}^- + 2h\, \mathfrak{a}_k^+ \mathfrak{a}_k^- + 2\bar{h}\, \mathfrak{a}_{k+1}^+ \mathfrak{a}_{k+1}^- \right)$$

$$+ t(-\mathfrak{a}_0^+ + \mathfrak{a}_0^-)(\mathfrak{a}_1^+ + \mathfrak{a}_1^-) - t(-\mathfrak{a}_L^+ + \mathfrak{a}_L^-)(\mathfrak{a}_{L+1}^+ + \mathfrak{a}_{L+1}^-) + (L-1)(f - h - \bar{h}), \quad (27)$$

where $a$, $b$, $c$, $d$, $h$, $\bar{h}$, $t$ and $f$ are given by (7)-(9). It is well-established that this type of fermionic models can be written as follows [27]

$$M' = \sum_{k=0}^{L+1} \lambda_k \left( \mathfrak{c}_k^+ \mathfrak{c}_k^- - \frac{1}{2} \right) + (L-1)f, \quad (28)$$

where $\mathfrak{c}_k^+$ and $\mathfrak{c}_k^-$ are also fermionic creation and annihilation operators and are linear combinations of $\mathfrak{a}_k^{\pm}$:

$$\mathfrak{c}_k^\epsilon = \sum_{\ell=0}^{L+1} \sum_{\tau=\pm} (\phi_k^\epsilon)_\ell^\tau\, \mathfrak{a}_\ell^\tau \quad \text{for} \quad k = 0, 1, \ldots, L+1, \quad \epsilon = \pm. \quad (29)$$

We recall that the coefficient in front of the identity operator in (28) may be determined by comparing the trace of $M'$ given by (24) and (28).

We recall briefly the computation of the coefficients $(\phi_k^\epsilon)_\ell^\tau$ and of the one-particle energy $\lambda_k$ in the appendix A. We get for the one-particle energy

$$\lambda_k = \frac{1}{\cos(2\theta)} \left( 2\sqrt{pq} \cos\left( \frac{\pi k}{L} \right) - \frac{p+q}{2} \left( \cos(2\theta) + \frac{1}{\cos(2\theta)} \right) \right), \quad \text{for} \quad k = 1, \ldots, L-1 \quad (30)$$

$$\lambda_0 = 0, \qquad \lambda_L = \lambda_{L+1} = -\frac{|p-q|}{2} \tan^2(2\theta). \quad (31)$$

Let us emphasize that all the one-particle energies are chosen negative. Then, by using the value (9) of $f$ and this choice of $\lambda_k$, relation (28) becomes

$$M' = \frac{|p-q|}{2} \tan^2(2\theta) + \sum_{k=0}^{L+1} \lambda_k \mathfrak{c}_k^+ \mathfrak{c}_k^-. \quad (32)$$

As mentioned above, the spectrum of $M$ can be deduced from the one of $M'$: the eigenvectors and eigenvalues of $M$ are the ones of $M'$ with an odd number of excitations and by discarding the excitation with vanishing energy [3, 19]. Namely, the eigenvalues of $M$ are given by

$$\Lambda = \frac{|p-q|}{2} \tan^2(2\theta) + \sum_{\ell=1}^{r} \lambda_{k_\ell}, \quad (33)$$

where $r$ is odd, $0 < k_1 < k_2 < \cdots < k_r \leq L+1$ and $\lambda_k$ are given by (30) and (31). In this way, one finds the $2^L$ eigenvalues of $M$.

The eigenvalues of $M$ with one excitation of type $\mathfrak{c}_L^+$ or $\mathfrak{c}_{L+1}^+$ vanish: they correspond to the two stationary states of $M$ (one of them being the trivial stationary state given by the empty lattice). The eigenvalue with one excitation of type $\mathfrak{c}_1^+$ in the thermodynamic limit corresponds to the spectral gap $G$ and is given by [19]

$$G = \begin{cases} -\frac{p}{\cos^2(2\theta)} \left( \sqrt{\frac{q}{p}} - \cos(2\theta) \right)^2 & \text{if} \quad p > q \\ -\frac{q}{\cos^2(2\theta)} \left( \sqrt{\frac{p}{q}} - \cos(2\theta) \right)^2 & \text{if} \quad q > p \end{cases} \quad (34)$$

It is also established in [19] that there exists a phase transition when the gap vanishes. For example, for $q > p$, the gap vanishes for $\sqrt{\frac{p}{q}} = \cos(2\theta)$ (or $(\Delta + 1)p = q$) and the system is in a low-density phase for $\sqrt{\frac{p}{q}} < \cos(2\theta)$ and a high-density phase for $\sqrt{\frac{p}{q}} > \cos(2\theta)$.

## 4  Spectrum of the inhomogeneous model

Using methods similar to the ones presented in section 3, we want to find the spectrum of the Markovian matrix (13),(16) corresponding to the inhomogeneous model. As in the homogeneous case, the first step consists in dealing with the boundaries. Thus, instead of $M$ given by (13) and (16), we study

$$M' = \sum_{k=1}^{L_1-1} \widetilde{m}_{k,k+1}^{(1)} + \widetilde{m}_{L_1,L_1+1}^{\text{junc}} + \sum_{k=L_1+1}^{L_1+L_2-1} \widetilde{m}_{k,k+1}^{(2)} + t_1 S_0^x S_1^x - t_2 S_{L_1+L_2}^x S_{L_1+L_2+1}^x. \tag{35}$$

where $\widetilde{m}^{(i)}$ are given by (6) with $p$, $q$ and $\theta$ replaced by $p_i$, $q_i$ and $\theta_i$ and $\widetilde{m}^{\text{junc}}$ are given by (14) without the terms proportional to $\tau$ and $\bar{\tau}$. In relation (35), the matrix $S^x$ acting in the space 0 (resp. $L_1 + L_2 + 1$) is given by (10) and (11) with $\theta$ replaced by $\theta_1$ (resp. $\theta_2$). We have also used the notation $t_i$ standing for the function $t$ (9) where $p$, $q$ and $\theta$ are replaced by $p_i$, $q_i$ and $\theta_i$. In the following, the same trick is used for the functions $a$, $b$, $c$, $d$, $h$, $\bar{h}$ and $f$. As previously, the spectrum of the inhomogeneous Markov matrix $M$ is deduced from the one of $M'$ (see below).

Now, $M'$ can be mapped to a quadratic fermionic operator. The Jordan-Wigner transformation for the inhomogeneous case is given by

$$\mathfrak{a}_k^+ = S_k^+ \prod_{j=0}^{k-1} S_j^z \quad \text{and} \quad \mathfrak{a}_k^- = S_k^- \prod_{j=0}^{k-1} S_j^z, \quad \text{for} \quad k = 0, 1, \ldots L_1 + L_2 + 1 \tag{36}$$

but with $S_j^{\#}$ given by (10) and (11) with $\theta$ replaced by $\theta_1$ if $0 \leq j \leq L_1$ and by $\theta_2$ if $L_1 + 1 \leq j \leq L_1 + L_2 + 1$. Then, by introducing $\mathfrak{c}_k^+$ and $\mathfrak{c}_k^-$ which are also fermionic creation and annihilation operators given by a linear transformation similar to (29), we get

$$M' = \sum_{k=0}^{L_1+L_2+1} \lambda_k \left( \mathfrak{c}_k^+ \mathfrak{c}_k^- - \frac{1}{2} \right) + (L_1 - 1)f_1 + (L_2 - 1)f_2 + \psi. \tag{37}$$

The one-particle energies $\lambda_k$ are computed in the appendix B. We get

$$\lambda_0 = 0, \qquad \lambda_{L_1+L_2} = -\frac{|p_1 - q_1|}{2} \tan^2(2\theta_1), \qquad \lambda_{L_1+L_2+1} = -\frac{|p_2 - q_2|}{2} \tan^2(2\theta_2) \tag{38}$$

and the other $L_1 + L_2 - 1$ one-particle energies $\lambda_1, \lambda_2, \ldots, \lambda_{L_1+L_2-1}$ are the solutions of the following equation in $\lambda$:

$$\left( \lambda + \overline{Q} + \bar{p} + \bar{q} \right) U_{L_1-1}\left( \frac{\lambda - 2f_1}{2\mu_1} \right) U_{L_2-1}\left( \frac{\lambda - 2f_2}{2\mu_2} \right)$$
$$= \mu_1 \frac{\bar{p}}{p_1} U_{L_1-2}\left( \frac{\lambda - 2f_1}{2\mu_1} \right) U_{L_2-1}\left( \frac{\lambda - 2f_2}{2\mu_2} \right) + \mu_2 \frac{\bar{q}}{q_2} U_{L_1-1}\left( \frac{\lambda - 2f_1}{2\mu_1} \right) U_{L_2-2}\left( \frac{\lambda - 2f_2}{2\mu_2} \right), \tag{39}$$

where $U_L(\cos(x)) = \sin((L+1)x)/\sin(x)$ are the Chebyshev polynomials of the second kind, $f_i$ is given by (9) and $\mu_i = \frac{\sqrt{p_i q_i}}{\cos(2\theta_i)}$.

Let us deduce from the spectrum of $M'$, the spectrum of the inhomogeneous Markov matrix $M$ given by (13). Firstly, in [3], they proved that we must discard the vanishing one-particle energy $\lambda_0$. Secondly, they showed that only two different cases can occur: *(i)* the eigenvalues of $M$ are the ones of $M'$ with an odd number of excitations; *(ii)* the eigenvalues of $M$ are the ones of $M'$ with an even number of excitations. The case *(i)* is the one used in section 3 for the homogeneous model. For inhomogeneous model, we must choose between these two cases.

To know which cases we must use, we compute the vacuum energy of $M'$

$$\Omega = -\frac{1}{2} \sum_{k=0}^{L_1+L_2+1} \lambda_k + (L_1-1)f_1 + (L_2-1)f_2 + \psi. \tag{40}$$

If this vacuum energy vanishes, the spectrum of $M$ is obtained from an even number of excitations whereas if it is positive, it is obtained from an odd number of excitations. This statement is proved by knowing that all the one-particle energies are negative, that the Markov matrix $M$ has only negative or vanishing eigenvalues and that there are only the two possibilities *(i)* and *(ii)* presented above.

Although there are no analytical expressions for the roots of (39), their sum is associated to the coefficient in front of $\lambda^{L_1+L_2-2}$. Then, one gets that

$$\Omega = \begin{cases} \frac{q_2-p_2}{2}\Delta_2 & \text{for } \Delta_1(q_1-p_1) \geq \Delta_2(q_2-p_2) > 0 \\ 0 & \text{for } \Delta_1(q_1-p_1) \geq 0 \geq \Delta_2(q_2-p_2) \\ \frac{p_1-q_1}{2}\Delta_1 & \text{for } 0 > \Delta_1(q_1-p_1) \geq \Delta_2(q_2-p_2) \end{cases} \tag{41}$$

From the above results, we deduce that the spectrum of $M$ is given by an odd number of excitations if

$$\Delta_1(q_1-p_1) \geq \Delta_2(q_2-p_2) > 0 \quad \text{or} \quad 0 > \Delta_1(q_1-p_1) \geq \Delta_2(q_2-p_2), \tag{42}$$

and an even number of excitations if

$$\Delta_1(q_1-p_1) \geq 0 \geq \Delta_2(q_2-p_2). \tag{43}$$

## 5 Spectral gaps for two particular models

In this section, we compute the spectral gap for inhomogeneous models. To simplify the presentation, we restrict ourselves to two particular cases and we set also $L_1 = L_2 = L$.

### 5.1 Impurity

We want to study here the effect of a single impurity between two identical segments. Therefore, we set

$$p_1 = p_2 = p, \qquad q_1 = q_2 = q \quad \text{and} \quad \theta_1 = \theta_2 = \theta. \tag{44}$$

The rates at the junction are given by

$$\bar{p} = p + s, \qquad \bar{q} = q + s \quad \text{and} \quad \overline{Q} = \left(\frac{q+p}{2} + s\right)\tan^2(2\theta), \tag{45}$$

where $s \geq -\min(p,q)$ is a free parameter. In this case $m^{(1)} = m^{(2)} = m$ where $m$ is given by (4) and

$$m^{\text{junc}} = \begin{pmatrix} 0 & 0 & 0 & 0 \\ 0 & -(q+s)(\Delta+1) & p+s & p+s \\ 0 & q+s & -(p+s)(\Delta+1) & q+s \\ 0 & (q+s)\Delta & (p+s)\Delta & -p-q-2s \end{pmatrix}. \tag{46}$$

The homogeneous lattice of length $2L$ is recovered for $s = 0$.

As explained previously, to get the spectrum and therefore the spectral gap, we must solve equation (39). In the cases treated here, the parameters satisfied (42). Therefore, the spectrum of the Markov matrix is obtained with an odd number of excitations. In particular, the spectral gap is obtained by adding the largest one-particle energy (we recall that the one-particle energies have been chosen negative) to this vacuum energy. We present on figure 3 the spectral gap in terms of $s$ for $L = 60$ (*i.e.* a lattice of length 120), $q = 3$ and $p = 0.5$ and for different values of $\theta = 0.1$, 0.5, 0.6, 0.65. Let us notice that for these values of $p$ and $q$, the phase transition of the homogeneous model described in section 3 is for $\theta \simeq 0.575$. The crosses on the Y-axis of figure 3 stand for the values of the spectral gap of the homogeneous case ($s = 0$) computed previously (34) for $L \to +\infty$. We see that the finite size effects are negligible since the curves corresponding to a lattice of finite length go through these points.

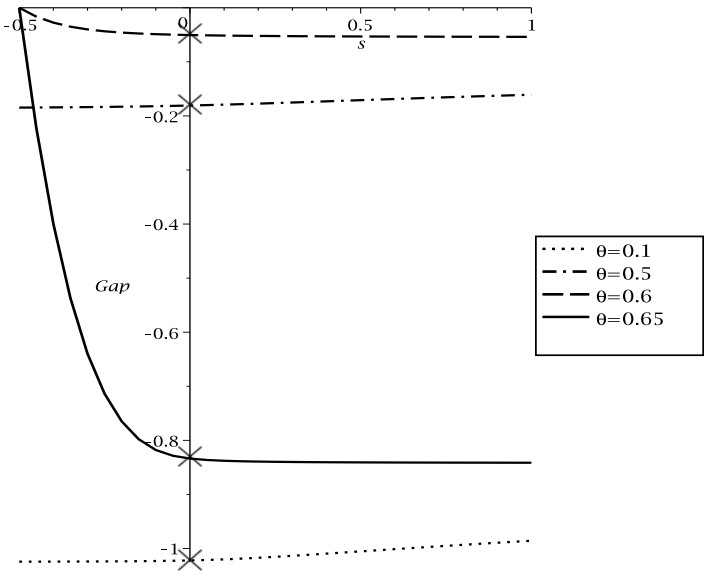

Figure 3: Spectral gap for the impurity model for $L = 60$, $q = 3$, $p = 0.5$.

For $\theta = 0.1$ or 0.5 and $s = 0$, the system is in a low-density phase (see section 3 and [19]). We see on the figure 3 that the introduction of the impurity has no significant influence on the spectral gap. For $\theta = 0.6$ or 0.65 and $s = 0$, the system is in a high-density phase. In this case, if the impurity slows down the particles (*i.e.* the rates at the impurity are smaller than the ones in the bulk, $s < 0$), the gap goes to zero. If at the impurity, the rates are greater than the ones of the bulk, the gap is almost unchanged. In summary, the impurity has only a significant influence in the high density phase when it is a slower junction than the ones in the bulk.

## 5.2 Spatial quench

In this subsection, we leave the rates of both segments free but we choose for the junction:

$$\bar{p} = p_1, \qquad \bar{q} = q_2 \qquad \text{and} \qquad \overline{Q} = \frac{p_1 \Delta_1 + q_2 \Delta_2}{2}. \tag{47}$$

In this case, the impurity jump operator becomes

$$m^{\text{junc}} = \begin{pmatrix} \frac{\Delta_2(q_2-p_2)-\Delta_1(q_1-p_1)}{2} & 0 & 0 & 0 \\ \frac{p_2\Delta_2-p_1\Delta_1}{2} & -\frac{q_1\Delta_1+q_2(\Delta_2+2)}{2} & p_1 & p_1 \\ \frac{q_1\Delta_1-q_2\Delta_2}{2} & q_2 & -\frac{p_1(\Delta_1+2)+p_2\Delta_2}{2} & q_2 \\ 0 & \frac{q_1\Delta_1+q_2\Delta_2}{2} & \frac{p_1\Delta_1+p_2\Delta_2}{2} & -p_1-q_2 \end{pmatrix}. \qquad (48)$$

In addition to $p_i, q_i, \Delta_i \geq 0$, the parameters must satisfy

$$\Delta_2 p_2 \geq \Delta_1 p_1 \qquad \text{and} \qquad \Delta_1 q_1 \geq \Delta_2 q_2, \qquad (49)$$

so that the rates of the impurity be positive. These rates are displayed on the Figure 4. Let us

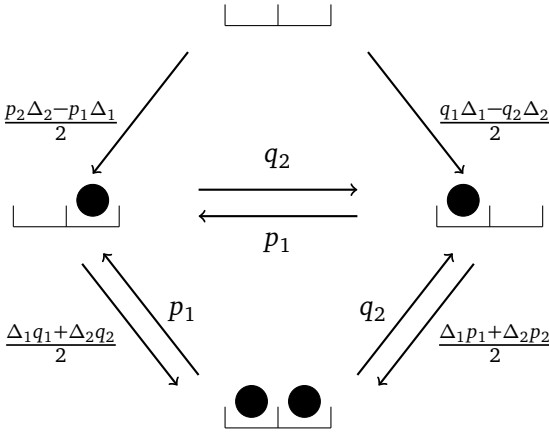

Figure 4: Nonvanishing rates between the different configurations at the junction.

remark that if the rates in both segments are identical then the impurity rates become equal to the bulk ones and we recover an homogeneous model.

We study in details the cases when $L = 60$, $p_1 = 0.6$, $q_1 = 6$, $p_2 = 6$ and $q_2 = 0.2$. For this case, the vacuum energy (41) vanishes and the spectrum of $M$ is obtained with an even number of excitations. Then, the spectral gap is the sum of the two largest one-particle energies. We plot in figure 5 the spectral gap w.r.t. $\Delta_2$ for different values of $\Delta_1$. We see on this figure that there are two different regimes. It corresponds to a crossing of the one-particle energies. For small $\Delta_2$, the two largest one-particle energies are $\frac{q_2-p_2}{2}\Delta_2$ and the largest solution of (39). For large $\Delta_2$, they are $\frac{p_1-q_1}{2}\Delta_1$ and still the largest solution of (39). The spectral gap depends greatly on the parameters of the second segment for $\Delta_2 \ll \Delta_1$ and of the first ones for $\Delta_2 \gg \Delta_1$.

## 6 Conclusion

We showed that inhomogeneous Markovian models can be mapped to free fermion models. We used this property to compute the spectral gap for different examples. We obtained that a local change of the rates may have a significant influence on the gap. Let us emphasize that the results and the methods used here can also be interesting to study quantum XY spin chains.

There are numerous open questions. For example, it is possible to compute for the inhomogeneous model other physical quantities like the density, the current or the correlation functions. To achieve this, it is necessary to generalize the methods developed to study the homogeneous case, like the empty interval method [14] or the matrix ansatz [21].

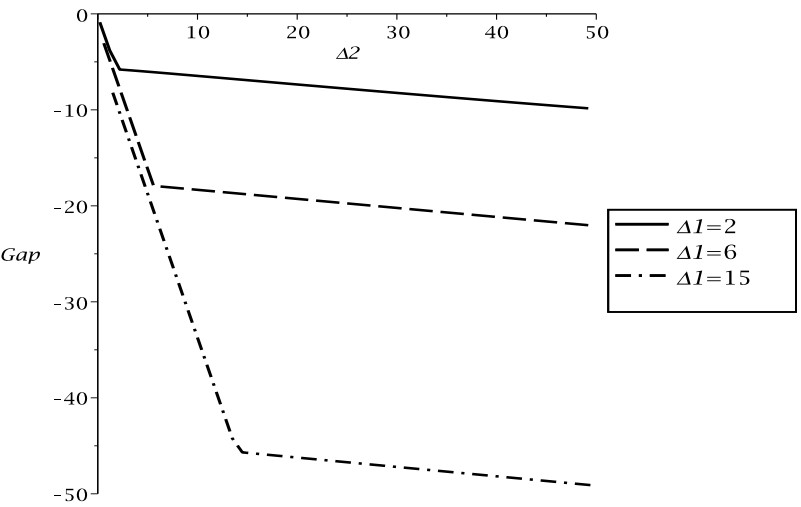

Figure 5: Spectral gap for the quench model for $L = 60$, $p_1 = 0.6$, $q_1 = 6$, $p_2 = 6$ and $q_2 = 0.2$.

There are other types of generalizations. There is a classification of the homogeneous Markovian models which can be mapped to free fermions [18]: they are four classes of such models. With methods similar to those presented in this paper, we can wonder if it is possible to pick up two different classes of models and glue them together. We can also try to study the cases with more than one impurity. Finally, we can also look for a Markovian model on a graph which can be mapped to fermionic models. Recently a XY model on a star graph was constructed in [9] where the Jordan-Wigner transformation was used to solve this problem.

**Acknowledgement:** I thank warmly V. Caudrelier, E. Ragoucy and M. Vanicat for their interests and their suggestions.

## A Computation of the one-particle energies $\lambda_k$ for homogeneous model

In this appendix, we compute the one-particle energies $\lambda_k$ and the coefficients $(\phi_k^\epsilon)_\ell^\tau$ of relations (28) and (29). By computing the commutator $[M', \mathfrak{c}_k^\epsilon]$ with $M'$ given by (27) or (28), we show that they must satisfy

$$\mathcal{M}\,\boldsymbol{\phi}_k^\epsilon = \epsilon\lambda_k\boldsymbol{\phi}_k^\epsilon \tag{50}$$

where $\boldsymbol{\phi}_k^\epsilon = (\,(\phi_k^\epsilon)_0^+, (\phi_k^\epsilon)_0^-, (\phi_k^\epsilon)_1^+, \ldots, (\phi_k^\epsilon)_{L+1}^-\,)^t$ and

$$\mathcal{M} = \begin{pmatrix} 0 & T & 0 & & & & & \\ T^t & H & I & 0 & & & & \\ 0 & J & H+\overline{H} & I & 0 & & & \\ & & & \ddots & & & & \\ & & 0 & J & H+\overline{H} & I & 0 & \\ & & & 0 & J & \overline{H} & -T & \\ & & & & 0 & -T^t & 0 & \end{pmatrix}, \tag{51}$$

and

$$H = 2\begin{pmatrix} h & \\ & -h \end{pmatrix}, \overline{H} = 2\begin{pmatrix} \overline{h} & \\ & -\overline{h} \end{pmatrix}, T = \begin{pmatrix} -t & -t \\ t & t \end{pmatrix}, I = \begin{pmatrix} -a & -c \\ d & b \end{pmatrix} \quad \text{and} \quad J = \begin{pmatrix} -b & c \\ -d & a \end{pmatrix}.$$
(52)

Then, we need now to find the $2L + 4$ eigenvectors and eigenvalues of $\mathcal{M}$.

Firstly, there exist two trivial eigenvectors with vanishing eigenvalue:

$$\boldsymbol{\phi} = \frac{1}{2} \, ( \, 1, 1, 0, \ldots, 0, 1, -1 \, )^t \qquad \text{and} \qquad \boldsymbol{\phi} = \frac{1}{2} \, ( \, 1, 1, 0, \ldots, 0, -1, 1 \, )^t.$$
(53)

Secondly, to find $L + 1$ other eigenvectors $\boldsymbol{\phi}$ of $\mathcal{M}$, we suppose that their components take the following form, for $\ell = 1, 2, \ldots, L$,

$$\begin{pmatrix} \phi_\ell^+ \\ \phi_\ell^- \end{pmatrix} = x^{\ell-1} \begin{pmatrix} 1+x \\ 1-x \end{pmatrix} - \left( \frac{p}{qx} \right)^{\ell-1} \begin{pmatrix} 1 + \frac{p}{qx} \\ 1 - \frac{p}{qx} \end{pmatrix}.$$
(54)

Then, the bulk part of (50) is satisfied if the eigenvalues are given by

$$\lambda = \frac{1}{\cos(2\theta)} \left( qx - \frac{p+q}{2} \left( \cos(2\theta) + \frac{1}{\cos(2\theta)} \right) + \frac{p}{x} \right).$$
(55)

Its boundary terms are satisfied if

$$\begin{pmatrix} \phi_0^+ \\ \phi_0^- \end{pmatrix} = \frac{1}{\lambda} \, T \begin{pmatrix} \phi_1^+ \\ \phi_1^- \end{pmatrix}, \qquad \begin{pmatrix} \phi_{L+1}^+ \\ \phi_{L+1}^- \end{pmatrix} = -\frac{1}{\lambda} \, T^t \begin{pmatrix} \phi_L^+ \\ \phi_L^- \end{pmatrix},$$
(56)

and if $x$ takes one of the following values:

$$x = \frac{p}{q} \cos(2\theta), \qquad x = \cos(2\theta), \qquad x = \sqrt{\frac{p}{q}} e^{i\pi\frac{k}{L}} \qquad \text{for} \qquad k = 1, \ldots, L-1.$$
(57)

The eigenvalues become respectively

$$\lambda = \frac{q-p}{2} \tan^2(2\theta), \qquad \lambda = \frac{p-q}{2} \tan^2(2\theta),$$
$$\lambda = \frac{1}{\cos(2\theta)} \left( 2\sqrt{pq} \cos\left( \frac{\pi k}{L} \right) - \frac{p+q}{2} \left( \cos(2\theta) + \frac{1}{\cos(2\theta)} \right) \right).$$
(58)

Finally, the $L + 1$ remaining eigenvectors are obtained starting from the following ansatz for the components of $\boldsymbol{\phi}$

$$\begin{pmatrix} \phi_\ell^+ \\ \phi_\ell^- \end{pmatrix} = x^{\ell-1} \begin{pmatrix} (1-x)\cos^4(\theta) \\ (1+x)\sin^4(\theta) \end{pmatrix} - r(x) \left( \frac{q}{px} \right)^{\ell-1} \begin{pmatrix} (1 - \frac{q}{px})\cos^4(\theta) \\ (1 + \frac{q}{px})\sin^4(\theta) \end{pmatrix},$$
(59)

where

$$r(x) = \frac{pq(x\cos(2\theta) - 1)(x - \cos(2\theta))}{(px\cos(2\theta) - q)(px - q\cos(2\theta))}.$$
(60)

By following the same steps as previously, one finds the eigenvalues given by (58) but with an opposite sign.

As explained in [3], we have a freedom in the choice of the sign for the one-particle energies $\lambda_k$. In this article, we choose the negative ones and we find finally that the one-particle energies are given by relations (30) and (31).

## B  Computation of the one-particle energies $\lambda_k$ for inhomogeneous model

In this appendix, we compute the one-particle energies $\lambda_k$ for the inhomogeneous model (37). They are determined by solving $\mathcal{M}\boldsymbol{\phi} = \lambda\boldsymbol{\phi}$ where

$$\mathcal{M} = \begin{pmatrix} 0 & T_1 & 0 \\ T_1^t & H_1 & I_1 & 0 \\ 0 & J_1 & H_1+\overline{H}_1 & I_1 & 0 \\ & & \ddots \\ & & 0 & J_1 & H_1+\overline{H}_1 & I_1 & 0 \\ & & & 0 & J_1 & N+\overline{H}_1 & A & 0 \\ & & & & 0 & B & H_2+\overline{N} & I_2 & 0 \\ & & & & & 0 & J_2 & H_2+\overline{H}_2 & I_2 & 0 \\ & & & & & & & \ddots \\ & & & & & & 0 & J_2 & H_2+\overline{H}_2 & I_2 & 0 \\ & & & & & & & 0 & J_2 & \overline{H}_2 & -T_2 \\ & & & & & & & & 0 & -T_2^t & 0 \end{pmatrix} \tag{61}$$

and $T_i$, $I_i$ $J_i$, $H_i$ and $\overline{H}_i$ are given by (52) with the functions replaced by the corresponding ones and

$$N = 2\begin{pmatrix} \eta & \\ & -\eta \end{pmatrix}, \quad \overline{N} = 2\begin{pmatrix} \bar{\eta} & \\ & -\bar{\eta} \end{pmatrix}, \quad A = \begin{pmatrix} -\alpha & -\gamma \\ \delta & \beta \end{pmatrix} \quad \text{and} \quad B = \begin{pmatrix} -\beta & \gamma \\ -\delta & \alpha \end{pmatrix}. \tag{62}$$

Now, we must find the eigenvalues $\lambda$ of $\mathcal{M}$ given by (61).

As previously, there exist two trivial eigenvectors similar than (53) with vanishing eigenvalues.

To obtain $L_1+L_2-1$ other eigenvalues, we suppose that the components of the eigenvectors $\boldsymbol{\phi}$ take the following form

$$\begin{pmatrix} \phi_\ell^+ \\ \phi_\ell^- \end{pmatrix} = \begin{cases} v\left[ x_1^{\ell-1}\begin{pmatrix} 1+x_1 \\ 1-x_1 \end{pmatrix} - \left(\frac{p_1}{q_1 x_1}\right)^{\ell-1}\begin{pmatrix} 1+\frac{p_1}{q_1 x_1} \\ 1-\frac{p_1}{q_1 x_1} \end{pmatrix} \right] & \text{for} \quad \ell = 1,2,\ldots,L_1 \\[2ex] x_2^{\ell-L_1-L_2-1}\begin{pmatrix} 1+x_2 \\ 1-x_2 \end{pmatrix} - \left(\frac{p_2}{q_2 x_2}\right)^{\ell-L_1-L_2-1}\begin{pmatrix} 1+\frac{p_2}{q_2 x_2} \\ 1-\frac{p_2}{q_2 x_2} \end{pmatrix} & \text{for} \quad \ell = L_1+1,\ldots,L_1+L_2 \end{cases} \tag{63}$$

The parameters $x_1$, $x_2$ and $v$ have to be determined to get an eigenvector. The parameter $v$ may be interpreted as a transmission factor. The boundary parts in 0, 1, $L_1+L_2$ and $L_1+L_2+1$ do not constrain these parameters. The bulk parts of the spectral problem give

$$\lambda = 2\mu_1 \cos(\zeta_1) + 2f_1 = 2\mu_2 \cos(\zeta_2) + 2f_2 \tag{64}$$

where

$$f_i = -\frac{p_i+q_i}{4}\left(1+\frac{1}{\cos^2(2\theta_i)}\right), \qquad \mu_i = \frac{\sqrt{p_i q_i}}{\cos(2\theta_i)} \qquad \text{and} \qquad \exp(i\zeta_i) = \sqrt{\frac{q_i}{p_i}}x_i. \tag{65}$$

New relations are given at the junction:

$$J_1\begin{pmatrix} \phi_{L_1-1}^+ \\ \phi_{L_1-1}^- \end{pmatrix} + (N+\overline{H}_1)\begin{pmatrix} \phi_{L_1}^+ \\ \phi_{L_1}^- \end{pmatrix} + A\begin{pmatrix} \phi_{L_1+1}^+ \\ \phi_{L_1+1}^- \end{pmatrix} = \lambda\begin{pmatrix} \phi_{L_1}^+ \\ \phi_{L_1}^- \end{pmatrix} \tag{66}$$

$$B\begin{pmatrix} \phi_{L_1}^+ \\ \phi_{L_1}^- \end{pmatrix} + (H_2+\overline{N})\begin{pmatrix} \phi_{L_1+1}^+ \\ \phi_{L_1+1}^- \end{pmatrix} + I_2\begin{pmatrix} \phi_{L_1+2}^+ \\ \phi_{L_1+2}^- \end{pmatrix} = \lambda\begin{pmatrix} \phi_{L_1+1}^+ \\ \phi_{L_1+1}^- \end{pmatrix} \tag{67}$$

which are equivalent to the following constraints between the parameters

$$v = \frac{\frac{1}{x_2^{L_2}} - \left(\frac{q_2 x_2}{p_2}\right)^{L_2}}{x_1^{L_1} - \left(\frac{p_1}{q_1 x_1}\right)^{L_1}} \tag{68}$$

and

$$\left(2f_1 + \overline{Q} + \bar{p} + \bar{q}\right)\sin(L_1\zeta_1)\sin(L_2\zeta_2) = \mu_1\left(\frac{\bar{p}}{p_1} - 1\right)\sin((L_1 - 1)\zeta_1)\sin(L_2\zeta_2)$$
$$+ \frac{\mu_2\bar{q}}{q_2}\sin(L_1\zeta_1)\sin((L_2 - 1)\zeta_2) \tag{69}$$
$$- \mu_1\sin((L_1 + 1)\zeta_1)\sin(L_2\zeta_2)$$

$$\left(2f_2 + \overline{Q} + \bar{p} + \bar{q}\right)\sin(L_1\zeta_1)\sin(L_2\zeta_2) = \mu_2\left(\frac{\bar{q}}{q_2} - 1\right)\sin(L_1\zeta_1)\sin((L_2 - 1)\zeta_2)$$
$$+ \frac{\mu_1\bar{p}}{p_1}\sin((L_1 - 1)\zeta_1)\sin(L_2\zeta_2) \tag{70}$$
$$- \mu_2\sin(L_1\zeta_1)\sin((L_2 + 1)\zeta_2)$$

Using relation (64), we can show that equation (69) implies equation (70). Finally, from (64) and (69), we get equation (39). The highest degree w.r.t. $\lambda$ in (39) is $L_1 + L_2 - 1$ then, solving it, one gets $L_1 + L_2 - 1$ eigenvalues.

Other $L_1 + L_2 - 1$ eigenvectors with the eigenvalues solution of (39) with opposite signs are obtained by starting from an ansatz for the eigenvectors similar than (59).

Two others eigenvectors are obtained from

$$\begin{pmatrix}\phi_\ell^+\\\phi_\ell^-\end{pmatrix} = \begin{cases} v_1 x_1^{\ell-1}\begin{pmatrix}1+x_1\\1-x_1\end{pmatrix} - v_2\left(\frac{p_1}{q_1 x_1}\right)^{\ell-1}\begin{pmatrix}1+\frac{p_1}{q_1 x_1}\\1-\frac{p_1}{q_1 x_1}\end{pmatrix} & \text{for} \quad \ell = 1, 2, \ldots, L_1 \\[4mm] x_2^{\ell-L_1-L_2-1}\begin{pmatrix}1+x_2\\1-x_2\end{pmatrix} - \left(\frac{p_2}{q_2 x_2}\right)^{\ell-L_1-L_2-1}\begin{pmatrix}1+\frac{p_2}{q_2 x_2}\\1-\frac{p_2}{q_2 x_2}\end{pmatrix} & \text{for} \quad \ell = L_1+1, \ldots, L_1+L_2 \end{cases} \tag{71}$$

with $x_1 = \frac{p_1}{q_1}\cos(2\theta_1)$ or $\cos(2\theta_1)$. They give the eigenvalues $\lambda = \pm\frac{q_1-p_1}{2}\tan^2(2\theta_1)$. The upper sign is for the former choice of $x_1$ and the lower sign for the latter. The parameter $x_2$ is constrained by the bulk equations and must satisfied

$$\frac{1}{\cos(2\theta_2)}\left(q_2 x_2 - \frac{p_2+q_2}{2}\left(\cos(2\theta_2) + \frac{1}{\cos(2\theta_2)}\right) + \frac{p_2}{x_2}\right) = \pm\frac{q_1-p_1}{2}\tan^2(2\theta_1). \tag{72}$$

The coefficients $v_1$ and $v_2$ are fixed by constraints (66)-(67) and are given explicitly by

$$v_1 = \frac{p_1\cos(2\theta_1)}{\bar{p}(p_1 - q_1 x_1^2)x_1^{L_1-1}x_2^{L_2}}\left[\left(\overline{Q} + \bar{p} + \bar{q} + \lambda - \frac{\bar{p}q_1 x_1}{p_1\cos(2\theta_1)}\right)\left(1 - \left(\frac{q_2 x_2^2}{p_2}\right)^{L_2}\right)\right.$$
$$\left. - \frac{\bar{q}x_2}{\cos(2\theta_2)}\left(1 - \left(\frac{q_2 x_2^2}{p_2}\right)^{L_2-1}\right)\right], \tag{73}$$

$$v_2 = \left(\frac{q_1 x_1^2}{p_1}\right)^{L_1}v_1 - \left(\frac{q_1 x_1}{p_1}\right)^{L_1}\frac{1}{x_2^{L_2}}\left(1 - \left(\frac{q_2 x_2^2}{p_2}\right)^{L_2}\right). \tag{74}$$

Finally, the last two eigenvectors are given by

$$\begin{pmatrix}\phi_\ell^+\\\phi_\ell^-\end{pmatrix} = \begin{cases} x_1^{\ell-1}\begin{pmatrix}1+x_1\\1-x_1\end{pmatrix} - \left(\frac{p_1}{q_1 x_1}\right)^{\ell-1}\begin{pmatrix}1+\frac{p_1}{q_1 x_1}\\1-\frac{p_1}{q_1 x_1}\end{pmatrix} & \text{for} \quad \ell = 1, 2, \ldots, L_1 \\[4mm] w_1 x_2^{\ell-L_1-L_2-1}\begin{pmatrix}1+x_2\\1-x_2\end{pmatrix} - w_2\left(\frac{p_2}{q_2 x_2}\right)^{\ell-L_1-L_2-1}\begin{pmatrix}1+\frac{p_2}{q_2 x_2}\\1-\frac{p_2}{q_2 x_2}\end{pmatrix} & \text{for} \quad \ell = L_1+1, \ldots, L_1+L_2 \end{cases} \tag{75}$$

with $x_2 = \frac{p_2}{q_2}\cos(2\theta_2)$ or $\cos(2\theta_2)$. They give the eigenvalues $\lambda = \pm\frac{q_2-p_2}{2}\tan^2(2\theta_2)$. The upper sign is for the former choice of $x_2$ and the lower sign for the latter. The parameter $x_1$ is constrained by the bulk equations and must satisfied

$$\frac{1}{\cos(2\theta_1)}\left(q_1 x_1 - \frac{p_1+q_1}{2}\left(\cos(2\theta_1) + \frac{1}{\cos(2\theta_1)}\right) + \frac{p_1}{x_1}\right) = \pm\frac{q_2-p_2}{2}\tan^2(2\theta_2). \qquad (76)$$

The coefficients $w_1$ and $w_2$ are fixed by constraints (66)-(67) and are given explicitly by

$$w_1 = \frac{q_2\cos(2\theta_2)x_1^{L_1}x_2^{L_2+1}}{\bar{q}(p_2-q_2 x_2^2)}\left[\left(\overline{Q}+\bar{p}+\bar{q}+\lambda - \frac{\bar{q}p_2}{x_2 q_2\cos(2\theta_2)}\right)\left(\left(\frac{p_1}{q_1 x_1^2}\right)^{L_1}-1\right)\right.$$
$$\left. -\frac{\bar{p}}{x_1\cos(2\theta_1)}\left(\left(\frac{p_1}{q_1 x_1^2}\right)^{L_1-1}-1\right)\right], \qquad (77)$$

$$w_2 = \left(\frac{p_2}{q_2 x_2^2}\right)^{L_2}w_1 - \left(\frac{p_2}{q_2 x_2}\right)^{L_2}x_1^{L_1}\left(\left(\frac{p_1}{q_1 x_1^2}\right)^{L_1}-1\right). \qquad (78)$$

As explained in [3] for the homogeneous case and previously in appendix A, we have a freedom in the choice of the sign for the one-particle energies $\lambda_k$. In this article, we choose again the negative ones and finally, one gets the one-particle energies (38)-(39).

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
