# Peer review of "Analytical results for a coagulation/decoagulation model on an inhomogeneous lattice"

_SciPost Physics, doi:SciPost Phys. 2, 006 (2017)_

## Round 1 · Referee Report · Anonymous (Referee 1) · 2017-1-26

Strengths

It is not simple to find inhomogeneos models that has a free fermion form. The solution they present (although not knew) give a good example of solving free fermion problems with boundaries.

Weaknesses

The english is just reasonable (mine is also!)

Report

The time evolution operator (Hamiltonian) of 1-d stochastic models corresponding
to the asymmetric diffusion of hardcore
particles together with coagulation and decoagulation are well known
(reference [1]) to be a free fermion one. In the present paper the author
verify that even in the case where we have a "link" defect joining two
homogeneous free fermions (with distinct process rates), can still be ruled
by a free fermion Hamiltoniam if the interaction is of special kind. Due
to the free fermionic nature of the inhomogeneous Hamiltonian they were able
to calculate their eigenpectrum, in particular the gap. They show, using
a numerical evolution, that in the
case where the defect connect two identical regions (left and right region with
the same parameters) the defect produce physical relevant consequence only
if the connect regions are in the dense phase, which is what we do expect.
The other application they did, is to fix a particular form for the
impurity interaction and ask for the forms of the two homogeneous Hamiltonian (l
eft and right of the defect) that would give a total free fermion
Hamiltonian.

The results presented are interesting. The solution method although not
knew (a generalization of the Bariev+Peshel ideas), produce an illustration of s
olution of a free fermion solution with
boundary terms.

It would be interesting if the author could provide a physical explanation
of their second application (Fig. 5) similarly as the one done in the
first application (Fig. 3).

Requested changes

1)It would be interesting if the author could provide a physical explanation of their second application (Fig. 5) similarly as the one done in the first application (Fig. 3). 2) Change: "...probability $qdt$..." " ...to probability proportional to $qdt$..." in the second paragraph of section 1.1. 3)The title of section 3, I think is missing "... the inhomogeneous.."

---

## Editorial Decision

published